# Flat-Faced or Non-Flat-Faced Cats? That Is the Question

**DOI:** 10.3390/ani13020206

**Published:** 2023-01-05

**Authors:** Greta Veronica Berteselli, Clara Palestrini, Federica Scarpazza, Sara Barbieri, Emanuela Prato-Previde, Simona Cannas

**Affiliations:** 1Department of Veterinary Medicine and Animal Science, University of Milan, 26900 Lodi, Italy; 2Department of Pathophysiology and Transplantation, University of Milan, 20133 Milan, Italy

**Keywords:** animal welfare, brachycephalic cats, cat behaviour, human-animal bond, inherited disorders, owner motivations, pet ownership

## Abstract

**Simple Summary:**

Despite the serious health and welfare issues related to brachycephalic breeds (e.g., pug, French bulldog, Persian, and exotic shorthair), their popularity is exponentially increasing. Brachycephalic means “shortened head” and refers to the short nose, and for this reason, the breeds with this morphological characteristic are called flat-faced breeds. The aim of this study is exploring differences in human-cat bonding, behavioural traits, perception of clinical breed-related problems, and the motivation for purchasing between brachycephalic cat owners (BCOs) and non-brachycephalic cat owners (NBCOs). This study aims to provide new knowledge on these topics, poorly explored in cats. As for owners of brachycephalic dogs, the character and appearance are the main motivations for purchasing these breeds. Most BCOs were not aware of several breed-related disorders (e.g., respiratory or ocular diseases) before acquiring their cat. Ownership of brachycephalic cat breeds results in being a complex and multidimensional phenomenon. A combination of motivations to own a brachycephalic cat and emotional engagement may explain this phenomenon.

**Abstract:**

Persian and exotic shorthair cats are the most-popular brachycephalic breeds worldwide. This study aimed to explore differences in human-cat bonding, behavioural traits, perception of clinical breed-related problems, and the motivation for purchasing between brachycephalic cat owners (BCOs) and non-brachycephalic cat owners (NBCOs). Using an online questionnaire, human-cat bonding and cats’ behavioural traits were explored using the CORS and Fe-BARQ scales, respectively. Breed-related problems and motivations for purchasing were explored only in BCOs. There were 278 BCOs and 250 NBCOs who completed the survey. Respiratory diseases resulted in being the main health problem of these breeds. Most BCOs were not aware of the incidence of these disorders in brachycephalic cats. Character and appearance were the main motivations for purchasing these breeds. Perceived emotional closeness (EC), cat-owner interaction (COI), and perceived cost (PC) mean scores were significantly higher in BCOs. Playfulness and affection-/attention-seeking scores were significantly higher in NBCOs. BCOs reported a significantly higher score for separation-related behaviours. The higher scores in separation-anxiety-related behaviours and in the EC and COI scales could be related mainly to the high level of care these breeds require. The motivation for acquiring brachycephalic breeds for good companionship seems in contrast with the lower scores obtained from BCOs for playfulness and affection-/attention-seeking.

## 1. Introduction

Flat-faced or brachycephalic breeds have become increasingly fashionable in the last few decades. Their popularity is exponentially increasing, and there is a real “brachy boom” [1,2,3,4]. Analysing some cat breed registers of 2021 (The Governing Council of the Cat Fancy; Cat Fanciers’ Association), the brachycephalic feline breeds (i.e., British shorthair, Persian, exotic shorthair, Scottish fold) are in the top ten positions in the ranking of the most-registered breeds [5,6].

Persian and exotic shorthair cats, considered flat-faced breeds per antonomasia, are among the most-popular cat breeds [5,6,7]. Persian is one of the oldest-known cat breeds [7]. The modern type is very different from the original one as the brachycephaly is more exaggerated. The “Peke-faced” or “Ultra-type” Persians represent the more-extreme degree of brachycephalism due to selective breeding [8]. Currently the Cat Fanciers’ Association considers the “Peke-face” to be the modern Persian standard. The standard of the exotic shorthair is comparable. The breed standard provides a round and massive head, with great breadth of the skull and a round face with a round underlying bone structure. When viewed in profile, the prominence of the eyes is apparent and the forehead, nose, and chin appear to be in vertical alignment. The nose must be short, snub, and broad, with the “break” centred between the eyes (i.e., the nose crosses the line joining the two lower eyelids) [6].

Anatomically, the feline brachycephalic breeds are characterised by their rounded skull, dorso-rotation of the jaws, particularly the maxilla, and a drastic shortening of the nose and face. In the extreme feline subjects, the upper canines are in the horizontal position and there is a kinking of the nasolacrimal ducts [9,10].

Brachycephalic breeds are more prone to developing clinical disorders based on the well-known combination of hereditary abnormalities: early ankylosis in the basicranial epiphyseal cartilage of the skull, which leads to chondrodysplasia of the longitudinal axis of the skull [9,10,11]. While the length of the skeletal muzzle is reduced, there is often no corresponding decrease in the size of the soft tissue contained in the skull. This results in a constricted effect within the nasal cavity and partial obstruction of the pharynx and larynx [11,12]. A combination of upper airway abnormalities—stenotic nares, elongated soft palate (rarely seen in cats), and abnormal nasal turbinates—that cause partial obstruction to a dog’s breathing results in brachycephalic obstructive airway syndrome (BOAS). BOAS has been characterised and standardised better in the dog than in the cat [12,13,14].

Brachycephalic cats are frequently affected by ophthalmic, facial, dental, respiratory, neurological, dermatological, and reproductive problems [10,15,16].

Brachycephalic traits have been implemented by breeders in response to a continuous demand for this type of flat-faced subject, often extremising these traits (hypertypical subjects). Therefore, in the last few decades, there has been a “brachy boom” with the emergence of a real brachycephalic crisis [2]. In this scenario, the concern about brachycephalic welfare has increased, as well as the scientific research into brachycephalic health disorders and the psychological mechanisms underlying this fascination and buying decision [17,18]. So far, research has mainly focused on dogs, but there is increasing evidence that brachycephalic cats experience the same poor welfare condition [4,8,9,16,19,20]. Despite the evidence of health problems related to these breeds and the awareness campaigns and multi-stakeholder working groups addressing the issues of brachycephalic welfare, these canine and feline breeds remain, across the years, among the most-widespread breeds in the world [2,5,6,17,21].

The reasons and the motivations for acquiring these types of cats have been less explored than in dogs [9,22]. In dogs, the appearance and the perceived positive behaviours (i.e., good for children, good companion) were found as priorities in the choice of brachycephalic breeds [23]. Furthermore, the fashion trend plays an important role in the decision-making [24].

Moreover, owners of brachycephalic cats seem to be less likely to undertake an in-depth search before buying their cat: they do not consider management requirements or health problems associated with the breed as important aspects in their decision-making [22].

In light of the available evidence, it is essential to better explore the reasons for choosing a brachycephalic breed in order to understand why the popularity of these breeds is still growing [20,23,25,26] and why the implemented awareness campaigns, the published scientific evidence on several health and welfare problems, and the individual negative experience with the same breed are not sufficient to discourage people from purchasing these breeds, which are even defined as “agony breeds” [20,27].

The main aim of this study was to further investigate the issue of brachycephalic cat breeds, exploring the relationship between owners of brachycephalic cats (BCOs) (including Person and exotic shorthair) and owners of non-brachycephalic cats (NBCOs (including other breeds and non-purebred cats), and to evaluate the behavioural differences between brachycephalic cat breeds and non-brachycephalic breeds. A secondary aim was to explore owner’s perception of clinical brachycephalic-related problems and the motivation for purchasing these breeds.

## 2. Materials and Methods

### 2.1. Survey Design and Distribution 

A questionnaire, adapted from a similar survey carried out on brachycephalic dog owners [26], was created to meet the needs of cat owners and designed to gather information on their motivations in purchasing, their perceptions of breed-related problems, cat behaviour (Fe-BARQ), human-animal bonding (CORS questionnaire), and cat medical history. The questionnaire was administered to two groups of owners: Owners of brachycephalic cats (BCOs) (Persian and exotic shorthair cats) and owners of non-brachycephalic cats (NBCOs) (including other pedigree cats and non-purebred cats). The whole questionnaire included four sections. The first section contained general information on participants (both BCOs and NBCOs), age, gender, education, and other background experiences, which could be relevant in determining their responses (past or actual pet ownership), and on the home environment. The second section provided information about cats (sex, reproductive status, current age, and age at acquisition), the origin of the cats (breeder, cat shelter/rescue, family/friends, or stray), veterinary expenses, and expectations and level of commitment required in cat management. Moreover, all participants were asked their perception of health status, major disease incidence, and the suffering of the brachycephalic breeds. A dedicated sub-section was developed only for BCOs to explore the cats’ medical history through specific questions on common breed-related health disorders and on motivations underlying the acquisition of a brachycephalic cat.

The third section was aimed at assessing the human–cat relationship. Owners completed the “Cat-Owner Relationship Scale” (CORS). This section comprised 26 questions divided into three subscales with a 1–5 multiple-choice response format. The CORS is a part of the Cat-/Dog-Owner Relationship Scale (C/DORS) [28]:Perceived emotional closeness (EC, 11 questions): This subscale is composed of items related to social support, affectional bonding, psychological attachment, companionship, and unconditional love. High scores represent a better quality of the relationship;Cat-owner interactions (COI, 6 questions): This subscale reflects both general activities related to the physical care of the pet, as well as to more intimate activities, such as kissing, cuddling, and hugging. High scores represent a better quality of the relationship;Perceived cost (PC, 9 questions): This subscale includes items assessing negative aspects of the relationship, including the financial, social, and emotional costs of caring for a pet. High scores represent a lower quality of the relationship (i.e., higher perceived costs of owing their pets indicate a less-positive relationship).

The C/DORS scoring was carried out following the instructions provided by the authors [28]. Thus, each item was scored on a five-point scale (1 to 5). The items for the pet–owner interaction and perceived emotional closeness subscales were reverse scored, such that a higher score indicates better perceived relationship quality. To calculate the score for each subject for each subscale, we summed the scores and, then, divided by the number of items of the subscale; then, we calculated the subscale average for each group (BCOs and NBCOs).

The C/DORS is based on the same approach, and it is structured as the Monash Dog–Owner Relationship Scale (MDORS). It combines into one single scale all the items from the MDORS and the additional items from the Cat-Owner Relationship Scale (CORS), a scale specifically designed for cat owners, used in the current study [28,29]. The MDORS is a well-documented tool specifically developed to evaluate the perceived relationship between humans and their dog from the human perspective and to check the strength or impact of the human-animal relationship as experienced by the human [30]. Social Exchange Theory, a psychological theory specifying that human relationships are maintained only when the perceived costs and benefits are balanced, was the starting point in the development of this scale. Based on this theory, both positive and negative aspects of pet ownership were considered [29,31].

In the fourth part of the survey, the owner was asked to respond to questions on cat behaviour, i.e., selected parts of the Fe-BARQ [32]. The Fe-BARQ is an equivalent of the Canine Behavioral Assessment and Research Questionnaire (C-BARQ) developed by [33] as a tool for describing typical responses of pet dogs to common stimuli in their natural environment. The questionnaire aims at obtaining quantitative behavioural evaluations of pet cats from their owners. In this part of the questionnaire, each behavioural item consists of a series of 5-point ordinal rating scales (e.g., never, rarely, sometimes, usually, always) with which the behaviour had been observed in the recent past (i.e., in the last few months) [32,33].

The considered behaviours were:Playfulness/activity (14 questions): to evaluate the tendency to interact in a playful manner with objects, people, or other pets in the environment;Sociability (15 questions): to evaluate the level of comfort the cat has with familiar and unfamiliar adults and children in the home;Affection-/attention-seeking (2 questions): to evaluate the solicitation or seeking out of attention from household members;Trainability (3 questions): to evaluate the propensity to respond to commands and attentive behaviours.Separation-related behaviours (6 questions): to evaluate the display of anxiety-related signs or abnormal behaviour just prior to or when left alone or separated from the owner for a period of time;

The questionnaire created for the study was in Italian, was made available on the free online survey tool “Google Forms” (Google), and remained available online from March 2020 until March 2021. A convenience sampling was performed. Respondents were recruited by sending the questionnaire link via social networks (Facebook and WhatsApp). The exclusion criterion for participation in the survey was being under 18 years of age. Participants were instructed to complete the questionnaire as thoroughly as possible: however, if they had no experience with the behaviour described, they were given the option to select “non-applicable” or “not observed” in the Fe-BARQ section; these responses were treated as missing values in the statistical analyses.

The study was reviewed and approved by Ethics Committee (reference number 97/20) University of Milan, Italy. The patients/participants provided their written informed consent to participate in this study.

### 2.2. Statistical Analysis

Answers to the questionnaires were scored and analysed using IBM SPSS Statistics 27 (SPSS Inc., Chicago, IL, USA). Descriptive statistics were calculated to provide a general description of the two groups of cat owners. Pearson’s Chi-squared test with Bonferroni correction was used to investigate possible differences between groups relating to the characteristics of the sample (demographics of the owners and cats), owner expectation p, veterinary experiences, and owner perception of brachycephalic health.

The Mann–Whitney U-Test was used for non-normally distributed continuous * categorical data (CORS score * breed and Fe-BARQ score * breed). The three CORS subscales’ scores were analysed as continuous variables. Linear regression was used to determine which factor predicted the three CORS subscales scores.’ Thirteen variables were tested for their association with the three CORS subscales scores’ using separate linear regression models: cat demographics (breed, age, sex, age, and source of adoption); owner demographics (sex, family members); veterinary routine; owner expectation of breed versus reality of ownership (veterinary cost, activity level, overall behaviour); owner perception of brachycephalic health.

The data of Fe-BARQ were not normally distributed (D’Agostino et al., 1990; Royston, 1991), so a Kruskal–Wallis equality of population rank test was used to detect the presence of a significant association between the Fe-BARQ factor score and cat demographics (breed, age, sex, age, and source of adoption); cat clinical data; owner demographics (sex, family members); veterinary routine; owner expectation of breed versus reality of ownership (veterinary cost, activity level, overall behaviour).

## 3. Results

### 3.1. Section 1: Demographic Information

The sample totalled 528 respondents (*n* = 278 brachycephalic cat owners (BCOs); *n* = 250 non-brachycephalic cat owners (NBCOs)). Of the total sample, 90.3% of respondents (477/528) were female. Most respondents were aged 31–44 years (33.9%) (179/528), and their highest level of education was an undergraduate degree (49.2%) (260/528). Respondents mainly lived in apartments with a balcony (50.5%) (267/528) or a semi-detached or detached house (36.7%) (194/528), with only 12.6% living in apartments without outdoor access (67/528). There were 38.5% of the households that consisted of two people (202/525) and 34.1% including children (180/528). There were 88.3% of the responded who had owned a cat previously (466/528). The demographic data for NBCOs vs. BCOs are outlined in Table 1.

### 3.2. Section 2: Cat Demographics, Clinical History and Owner Expectation

A third of the cats (28.8%, 152/528) were aged between 1 and 3 years. In the study population, 44.7% of the cats were female (236/528) and 24.6% of them were intact; 55.3% were male (252/528), and 27.1% of them were intact. Regarding the non-brachycephalic group, 76.4% (191/250) were not purebred cats, and Maine Coon was the most represented pure breed (10.4%, 26/250). Most cats were adopted between two and four months (56.6%, 299/528), and 33.9% were acquired by pedigree breeding. The data are presented by breed in Table 2.

There were 46% (243/528) of the cats that underwent a veterinary check twice yearly, while 3.5% (19/528) were visited five times per year. Most owners believed that brachycephalic breeds suffer more than non-brachycephalic ones (61.7%, 326/528) and have more health problems than other cats (74.6%, 394/528). Considering only BCOs, the same result regarding the perception of brachycephalic cats suffering emerged, even if the percentage was lower (54.7%, 152/278). Considering all the owners (BCOs + NBCOs) believing that brachycephalic cats have more problems than others (394/528), respiratory diseases were considered the main health problem of brachycephalic breeds (82.2%, 324/394), followed by ocular problems (74.3%, 293/394), dermatological diseases (30.7%, 121/394), and behavioural problems (8.3%, 33/394). Considering BCOs versus NBCOs, the ocular problems were considered the main health problem by BCOs (50.7%, 141/278), while the respiratory diseases remained the main health problem for NBCOs (70.4%; 187/250). The satisfaction of most owners (BCOs + NBCOs) met the expectation for all the items considered: veterinarian expenses: 52.6% (278/528); commitment request 62.8% (332/528); interaction 54.7% (289/528); general behaviour 43.5% (230/528). The data presented in Table 3 show the difference between the two groups. 

#### Brachycephalic Cat Owners

The 75.1% (209/278) and 43.1% (120/278) of owners were not aware before acquiring their cat of breathing disorders (BOASs) and of other specific-breed-related health disorders (i.e., ocular, skin, heart, neurological, and reproductive diseases), respectively. Owners were asked to evaluate the health status of their cat, and half of the respondents (50.4%, 140/278) considered it excellent. The main motivation reported by owners for acquiring these breeds was the character (92.4%, 257/278) and the aesthetic appearance (87%, 242/278). Regarding the physical characteristics, 95.3% (265/278) of owners loved the big and round eyes, following by the rounded head (92.8, 258/278), the flat face (80.4%, 243/278), and the skin folds (36.7%, 102/178). The cuteness (98.2%, 273/278), the funny-lookingness (96.7%, 266/278), and the human-like face (78.4%, 218/278) were the main traits loved by the owners of these cat breeds. The cats’ health information reported by BCOs is summarised in Table 4.

### 3.3. Section 3: Cat Owner Relationship Scale

The CORS scores were high for all three sub-scales with means and medians >4.0 for EC, COI, and PC for both groups. The COI, EC, and PC means were significantly higher in BCOs than NBCOs (*p* ≤ 0.05) (Figure 1, Figure 2 and Figure 3).

Breed was associated with the EC, COI, and PC subscale scores in the linear regression (*p* ≤ 0.05); family members were associated only with COI, and veterinary routine resulted in being associated with PC (*p* ≤ 0.05). For the subscale PC, the overall regression was statistically significant (R^2^ = 0.28, F = 2.518, *p* = 0.021). For the subscale EC, the overall regression was statistically significant (R^2^ = 0.68, F = 2.884, *p* = 0.000). For the subscale COI, the overall regression was statistically significant (R^2^ = 0.50, F= 2.249, *p* = 0.009).

Analysing only brachycephalic breeds, we found that the age of the cat was associated with EC subscale scores and the source of adoption with PC subscale scores in the linear regression (*p* ≤ 0.05). In particular, the EC subscale score of cats aged under 6 months was found lower than the other classes of age, and the PC subscale score of cats adopted by certificated breeders was lower than the other cat sources.

### 3.4. Section 4: Fe-BARQ 

The Fe-BARQ mean score was significantly higher for playfulness (Z = −3.142; *p* = 0.002) and affection-/attention-seeking (Z = −3.035; *p* = 0.002) in the non-brachycephalic group than in the brachycephalic one (Figure 4 and Figure 5).

Separation-related behaviours’ mean scores resulted in being significantly higher in the brachycephalic group (Z = −2.087; *p* = 0.037) (Figure 6). 

Analysing only brachycephalic breeds, a high score of trainability was associated significantly with respiratory difficulty during the hot season (H = 10.734; *p* = 0.030) (Figure 7). Considering only Persian cats, they were significantly distinguished by a higher score of trainability (H = 9.749; *p* = 0.045). Considering separation-related behaviours, a high score was significantly associated with respiratory difficulty during and/or after feeding (H = 11.704; *p* = 0.020). 

## 4. Discussion

Ownership of brachycephalic breeds is a complex and multidimensional phenomenon and has been explored more deeply in dogs than in cats so far. It is characterised by a strong human-animal bond and by distorted perceptions of good health set against high levels of disease and poor welfare [25,26]. There is little research on cat brachycephaly linked to owner motivation and cat welfare [9,22], and to our knowledge, this is the first work in Italy that explores the phenomenon in brachycephalic cat breeds such as Persian and exotic shorthair cat breeds. Using an online questionnaire, we assessed brachycephalic and non-brachycephalic cat owners’ motivation for purchasing cats and perception of their cats’ health and welfare, as well as cat behaviour and the human-cat bond.

The majority of participating owners were female, with no significant difference between the two groups. The general profile of the brachycephalic cat owner emerged in this study was overall similar to that recently depicted by Plitman et al. [22], who found that owners of these breeds were mainly women, aged between 45 and 54, with a high level of education, married or in a relationship, with no children, and that already owned cats.

In our study, most respondents were women, aged between 31 and 54, with a middle and secondary school level of education, living in a family composed of two members, without children, and with previous experience of cat ownership. These findings are largely compatible with previous surveys on cat ownership [34,35] and confirm that women are more likely to respond to surveys and seem to be more involved in pet management and care [36,37,38]. Brachycephalic dog and cat ownership seems to differ with respect to the presence of children within the household because brachycephalic canine breeds are perceived as good with children, and this is one of the motivations in purchasing these dog breeds [23]. Moreover, in this study, owners without children in the household exhibited higher COI with their cats, similar to what was found by Packer et al. [26]. This might be due to the time constraints of parenting, not allowing for the same level of pet-owner interaction afforded to owners without children, and the emotional demands of parenthood can affect the level of emotional bonding with their pet [26,39].

Unlike what was reported for brachycephalic dogs, brachycephalic cat owners (BCOs) were older (age range 31–54 yrs.) than non-brachycephalic cat owners (NBCOs (age range 18–30 yrs.). Young people have been reported to be more influenced by social media to acquire a brachycephalic dog [23,26]. Although there is considerable visibility of brachycephalic cats in social media or advertisements, it is possible that brachycephalic cat breeds may not have the same appeal to young people as brachycephalic dog breeds.

In this study, the brachycephalic cat profile was that of a young cat (<3 years old), adopted between 2 and 4 months of age from a registered breeder. Unneutered male cats were preponderant. Our results differed from the literature regarding the cat’s gender, where females and males were equally represented and most cats were neutered [22,40,41]. This could be explained by the fact that many purebred cats are breeding subjects [22].

In addition, buying a pet from a registered breeder may presuppose a certain attention in the choice to have guarantees regarding the breed standard or health status. It can be assumed that the owners of brachycephalic cats acquired by registered breeders presumably would like to purchase an individual with specific physical and behavioural characteristics, and a subject coming from a registered breeder can meet these needs [22].

Owners of brachycephalic cats were less likely to undertake in-depth research before buying their cat and did not consider management requirements or health problems associated with the breed as important points in their decision-making [22].

Our findings suggest the existence of some significant differences between BCOs and NBCOs in going to the veterinarian annually, the perception of the level of suffering and the main clinical-related disorders, and level of satisfaction regarding their cats. Paradoxically, although 36% of BCOs visited the vet from 3 to 5 times per year (NBCOs 20%), the perception that brachycephalic breed were suffering more than other breeds was expressed significantly more by NBCOs (70%) than BCOs (55%) despite most of them claiming that their cat showed many symptoms of breed-related diseases (i.e., retching or vomiting after meals; snoring during sleeping; epiphora and noisy breathing; heat intolerance). It is also interesting to point out that the respiratory and ocular disorders were mostly considered as problems of brachycephalic breeds by NBCOs. Although most BCOs affirmed that they were not aware of these issues typical of the breed before adoption, dermatitis was the main veterinary diagnosis, but not significantly [4]. Pruritis and pain caused by dermatological disorders can have a severe impact on the quality of life of affected animals and their owners, due to the stress associated with the long-lasting therapy regimen, non-easy treatment administration in cats, and visits to the veterinarian [42,43,44]. This can also be reflected in the higher score of perceived cost (PC) and increased frequency of access to veterinary care shown by BCOs. Compared to other breeds, Persians have an increased risk of coat and skin problems (e.g., dermatophytosis and pseudomycetomas), which may be associated with their dense and long coats and with a genetic predisposition. Moreover, irregular grooming can cause severe matting in longhair cats with potentially negative dermatological consequences; Persian cats not properly trained from an early age can become aversive to being groomed with possible negative health and behaviour consequences [45,46].

Although no significant differences between groups emerged in this study, for most BCOs, the satisfaction with veterinary costs, commitment, and interaction paradoxically met their expectations. These results seem in contrast with those of Plitman et al. [22], who reported that BCOs underlined significant care and maintenance issues (i.e., long hair grooming and eye cleaning) and, for these reasons, were less likely to recommend their cat’s breed to others.

Unlike Plitman et al. [22], who found that BCOs were less likely to consider their cat as healthy, in the present study, the majority of BCOs considered their cat’s health to be excellent despite the onset of health problems or the need to deal with the veterinarian more frequently, resulting in an increase in costs and efforts in caring [22]. However, as mentioned above, the perceived cost (PC) by BCOs was significantly higher than that of NBCOs. Interestingly, these results are consistent with those reported by [26] on brachycephalic dog breeds.

Despite this, owners did not seem to regret having bought a brachycephalic pet, but rather, were satisfied with the choice made. The explanation for this would be that inherited disorders and morphology-related problems are considered as “normal for the breed” by BCOs and, thus, accepted and, sometimes paradoxically, desired [20,26,27,47]. The owner of a brachycephalic dog expects from his/her pet a minimal need for activity and the expression of some “anomalous” behaviour (e.g., snoring or panting during the hot season). Packer et al. [47] found that most owners of brachycephalic dogs with clinical signs of BOAS did not consider that their pets suffered from a specific respiratory disease, but rather often justified these signs as normal for the breed. Moreover, when owners of brachycephalic dogs were asked to compare their own dog’s health status to the rest of their breed, a concerning trend towards owner over-evaluation of the dog’s relative health emerged [26]. A cognitive dissonance process has been hypothesised to explain this deflection phenomenon: owners are aware of their dog’s breed-related health problems, “but find accepting these problems in their own dog as psychosocially uncomfortable, instead deflecting the issue to other individuals” [26]. It was also suggested that a psychological conflict could be linked to this phenomenon: owning and loving an individual of a breed with several well-known health problems at the breed level [26,47].

Considering the contrasting and even paradoxical results of our study, it is plausible that a cognitive dissonance process could be involved. BCOs could be aware that their cat shows a number of clinical symptoms, and considering them as “normal traits” of the breed somehow acceptable [47] or overestimating their cats’ health could allow them to overcome the psychological discomfort associated with buying an inherently unhealthy breed.

Cognitive dissonance theory postulates that an internal psychological tension/conflict arises when an individual’s behaviour is inconsistent with his/her thoughts, beliefs, and feelings [48]. The individual is therefore motivated to reduce the conflict by justifying his/her decision, and the justification is typically achieved by changing attitudes and beliefs, making them consistent with the decision/behaviour that has been made [49,50]. Thus, understanding people’s perception and attitudes toward brachycephalic breeds is fundamental to effectively address the cognitive and emotional dissonance that seems to characterise the human–brachycephalic cat relationship [26,47].

The satisfaction with their behaviour differed significantly between the two groups, as for BCOs, it was better than expectations. The behavioural traits of brachycephalic breeds are some of the motivations for purchasing them. These breeds are perceived as good companion pets because they are friendlier and more affectionate and have lower needs [22,26,43]. The perception of brachycephalic breeds’ positive behavioural traits is supported by some empirical evidence focused on brachycephalic dogs. Dogs with shorter muzzles (higher cephalic index) were reported to be more affectionate, cooperative, and interactive with humans than dogs with longer muzzles (lower cephalic index) [51,52]. Some studies suggest that brachycephalic dogs have a better performance in utilising the human point gesture and in looking at human and dog portraits compared to dolichocephalic and mesocephalic breeds [53,54,55]. This could explain the popularity of these breeds, as humans could prefer a pet that looks at them longer, giving the perception of better interaction and communication [56]. Recently, however, it was suggested that breeds with exaggerated facial conformations, such as exotic shorthairs and Persians, have more limited abilities to produce clearly identifiable and differentiable facial expressions [57,58]. These limitations can have a relevant impact on the expression of emotions, pain, and intentions with consequences on social relations both with humans and conspecifics [59,60,61,62].

To deepen how breed-linked morphology might impact the behavioural repertoires, the cat’s behavioural traits were explored through the Fe-BARQ. A lower score of playfulness and affection-/attention-seeking characterised brachycephalic cats. In contrast, non-brachycephalic cats were significantly more affectionate, interactive, and playful. These results are partially consistent with previous studies that reported Persians as less active and playful than other breeds, such as Siamese [39,63,64], and less friendly and affection-seeking [65]. Our findings seem not to fully support the assumption that brachycephalic cat breeds are more affectionate or interact better with their owner. Additionally, regarding trainability, we found that Persians were more trainable. The perception of a high level of trainability could be explained by the craniofacial conformation, which may give rise to the impression that brachycephalic pets are more engaged with their owners, as previously mentioned [56]. Further investigations are needed to confirm these results. It is important, however, to consider that the survey was carried out during COVID restrictions. Thus, possible changes in daily routine and/or in management could have negatively affected the cats’ behaviours (e.g., onset or deterioration of anxiety-related disorders [66,67], or positively, improving welfare due to the possibility of increased human-cat interaction [68].

The significantly higher score in separation-related behaviours shown by brachycephalic cats confirms the finding that these breeds, Persians in particular, are more likely to suffer from separation anxiety [69]. The occurrence of separation-related problems in domestic cats has been reported by a few studies [39,70,71]. Separation anxiety is defined as a set of behaviours and physiological signs displayed by the pet when separated from its attachment person, and it has been well studied in the dog [69,72]. Hyper-attachment to the owner was significantly associated with separation anxiety in the dog, but it has not been proven in the cat [73]. Brachycephalic breeds have been selected for a neotenous appearance, favouring consequently neotenous behaviours, such as attachment and behavioural dependency, to satisfy specific human needs [52,74]. The correlation between the high score of separation-related problems and the presence of respiratory difficulties during/after feeding can underline that the onset of disease or clinical symptoms may contribute to the attachment process by acting as releasers of parental nurturing and caregiving behaviour [25,75]. One explanation for this correlation could be that owners of animals showing clinical disorders become more attached to their pets due to the extra care and protection they are perceived to need [25]. The study of Kurdek in dogs reported that a high level of caregiving is associated with a high level of attachment to the animal; we can therefore reasonably think that it also happens in cats [76]. Thus, it is possible that the poor health condition of companion animals, and the consequent caregiving requirements, is a desirable trait for some owners as it satisfies their caregiving need. It was pointed out that a strong emotional bond between owners without children (particularly women) and their dogs can reflect the caregiver-infant relationship of owners seeking “perpetual children” [26,77]. The dependency of dogs on their owners did not seem to be perceived as a drawback, but rather, the owners seemed to bask in this situation and feed it [27,78]. The same cannot be ruled out for cats: cats in general are much more independent of the owners than dogs, but the brachycephalic cats with their fragility represent an excellent opportunity to establish this type of bonding.

The current study confirmed that the aesthetic appearance, as well as the behavioural traits are the main motivations for acquiring brachycephalic breeds (i.e., Persian and exotic shorthair) [22,23,42]. Health and longevity issues have a lower influence on the owners that choose to keep brachycephalic dogs [23,25]; big and round eyes and rounded head were the more attractive physical characteristics. At the emotional level, cuteness and funny-lookingness were the main traits appreciated by the BCOs involved in this study. This can be explained by the “baby-schema effect” (“*Kindchenschema*”), which was initially described by Konrad Lorenz [20,79]. This “baby-schema effect” is based on infantile features such as round cheeks, a large head, big eyes, a high forehead, thick and short extremities, and clumsy movement, which are perceived as cute and trigger an emotional reaction and a caring response in adults [68,80] and are preferred since infancy [81]. The “baby-schema effect” is one of the well-recognised mechanisms underlying the popularity of brachycephalic breeds [20,82,83]. The “brachy boom” has caused an evident growth of ill-considered breeding by makeshift and commercial breeders feeding an illegal international puppy trade in response to an increasingly high demand for subjects with hypertypical characteristics [2]. Appearance-driven breeding is acknowledged to be problematic for the health and welfare of pets. Thus, it is alarming that appearance is one of the primary motivations in owners’ purchasing [2,11,19,42].

The attraction of infant characteristics in companion animals can potentially influence the human-animal bond, and there is evidence that the degree of attachment reported by the owners is associated with their rating of the overall attractiveness of dogs and cats showing infant features. [31,74]. Moreover, dogs perceived as cute by owners were perceived to also be friendly, safe, and affectionate [84]. These findings support the hypothesis that the owner’s interpretation of canine behaviour and personality may be influenced by the dog’s appearance, such as the infantile features typical of brachycephalic breeds [81].

The results of the current study on the cat-owner relationship are in line with those found in brachycephalic dog breeds (e.g., French and English bulldogs and pugs), and they indicate that also BCOs form a strong relationship with their pet [23,26,30]. A high score of the EC subscale represents the high quality of relationship related to social support, affectional bonding, psychological attachment, companionship, and unconditional love. However, these findings could be influenced by the prevalence of women among the respondents, since they generally exhibit a stronger attachment and emotional bonding to companion animals than men [6,85,86]. This is also confirmed by the high scores in the COI subscale, which reflect both activities related to the physical care of the pet, as well as to more intimate activities (i.e., kissing, cuddling, and hugging). The intensity of the relationship and the emotional involvement seem to depend also on the age of the cat. The younger the cat, the lower the score given on the EC scale is. This could be due to the fact that the human–animal bond may develop and strengthen with age and experience [39]. In this case also, our results may have been affected by the restrictions imposed by the pandemic situation. There is evidence that pets provided owners with substantial emotional support to mitigate the negative effects of confinement, and this could have influenced the responses [66,68].

In addition, the important role played by social influence (fashions and fads) and media exposure (e.g., featuring in movies, pets of VIP, social profiles dedicated to companion pets) in the popularity of companion pet breeds should not be overlooked, especially for dog breeds (e.g., French bulldog and pug) [23,24,32].

In addition, brachycephalic breeds are perceived as suitable to the modern lifestyle needs of a companion pet: they fit with the reduction of living space (i.e., apartments), the busy lifestyle, city-living, and households [23]. These aspects remain however unexplored in the brachycephalic feline breeds, but it is speculated that they can amplify these breeds’ popularity [23,38].

The study has a number of limitations. On the one hand, the use of a convenience sample is not representative of the Italian cat owner population. Moreover, as has already emerged in these kinds of studies [36,37,38], most participants were women. Future studies should try to recruit more male respondents in order to minimise sampling bias related to the owner’s gender. Furthermore, the survey was a self-selected one: people chose to participate, and this introduces a systematic error due to a non-random sample of the population [87,88]. It was administered online, thus limiting access only to respondents who had an Internet connection and were comfortable with this technology. Finally, being a self-report, information on the cat–human relationship, cat behaviour, and motivation to purchase might have been influenced by the way in which the subjects were interpreting the questions. Additionally, some owners could have been unwilling to admit, for example, the presence of clinical issues for their pet and their real emotions or perceptions regarding cat ownership [41].

The current study represents a snapshot of Italian cat owners at a particular point of the COVID restrictions [66,67,68]. It is a descriptive and correlational study exploring some aspects of the ownership of brachycephalic cats. Despite its limitations, these findings are a starting point for further research addressing the complex issue of brachycephalic breed ownership.

## 5. Conclusions

Much of the recent literature concerning brachycephalic breeds’ health and welfare and owners’ attitudes has focused mainly on dogs. Despite growing evidence that brachycephalism in cats is directly associated with disorders that negatively affect their health and welfare conditions, brachycephalic breeds continue to be very popular, leading to a real brachycephalic crisis. The current study expands the knowledge on the complex phenomenon of brachycephalic breeds’ ownership, exploring the feline component. The results are in line with the findings from canine research. The cat’s physical appearance and the perception of its behavioural traits appear to be the main motivations for purchasing these breeds. As for brachycephalic dogs (i.e., pug, English and French bulldogs), the presence of clinical problems does not discourage people from keeping them. The strength of the human-animal bond increases the likelihood of acquisition, since a high emotional closeness can balance the perceived costs of the ownership. This feeds the desire to continue owing these breeds.

As for brachycephalic dogs, a cognitive dissonance process could to be involved with brachycephalic cat owners under-recognising or underestimating the poor welfare and health problems of their cats to overcome the psychological discomfort associated with buying an inherently unhealthy breed. Understanding people’s perception and attitude toward brachycephalic breeds is crucial to implementing adequate strategies to solve the cognitive and emotional dissonance characterising this type of pet–human relationship, to mitigate breed-related welfare problems, and finally to address the brachycephalic crisis.

## Figures and Tables

**Figure 1 animals-13-00206-f001:**
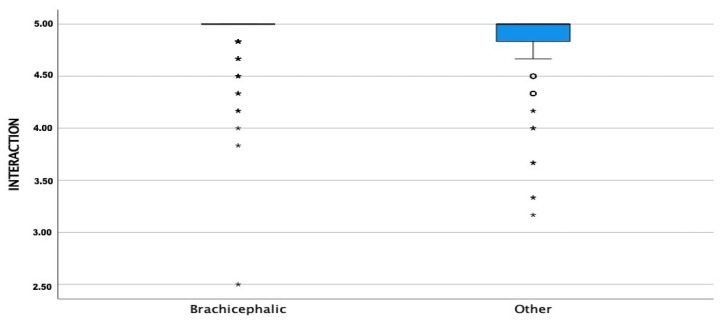
Comparison of cat-owner interaction subscale (COI) means between brachycephalic and non-brachycephalic breeds. Dots represent weak outliers; stars represent strong outliers.

**Figure 2 animals-13-00206-f002:**
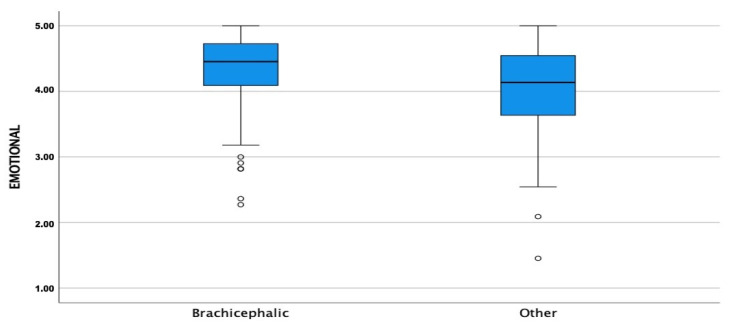
Comparison of emotional closeness subscale (EC) means between brachycephalic and non-brachycephalic breeds. Dots represent weak outliers.

**Figure 3 animals-13-00206-f003:**
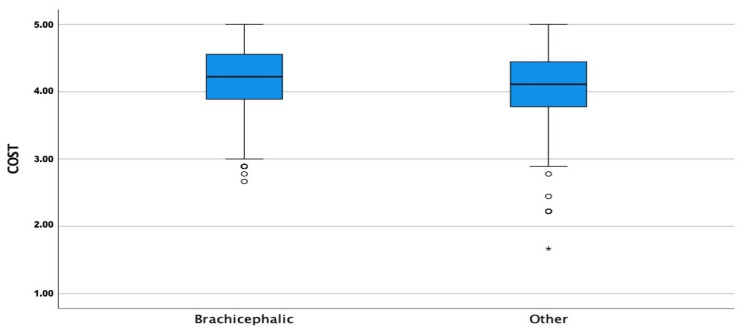
Comparison of perceived cost subscale (PC) means between brachycephalic and non-brachycephalic breeds. Dots represent weak outliers; stars represent strong outliers.

**Figure 4 animals-13-00206-f004:**
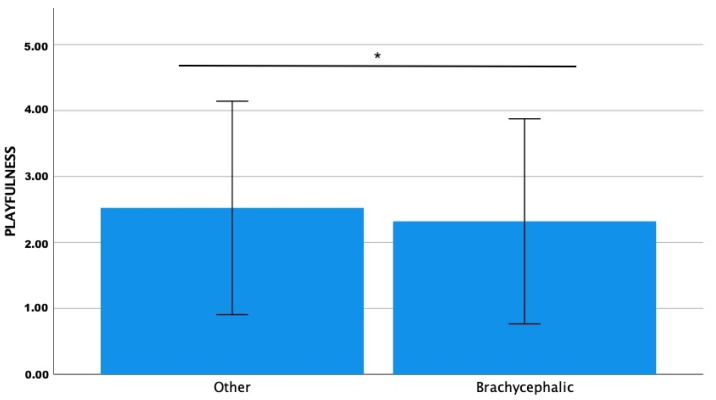
Comparison of Fe-BARQ mean scores for playfulness between brachycephalic and non-brachycephalic breeds. * *p* = 0.002.

**Figure 5 animals-13-00206-f005:**
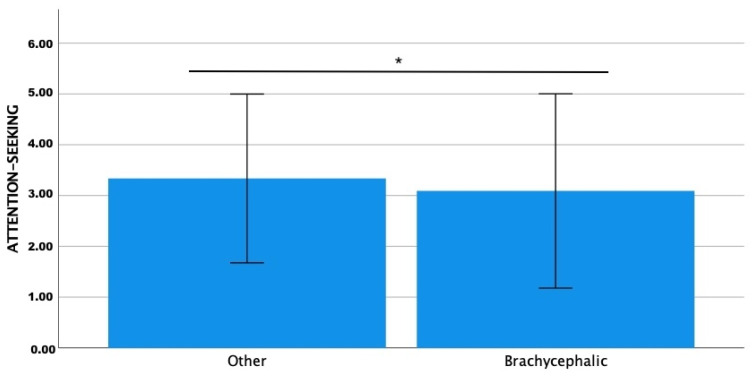
Comparison of Fe-BARQ mean scores for affection-/attention-seeking between brachycephalic and non-brachycephalic breeds. * *p* = 0.002.

**Figure 6 animals-13-00206-f006:**
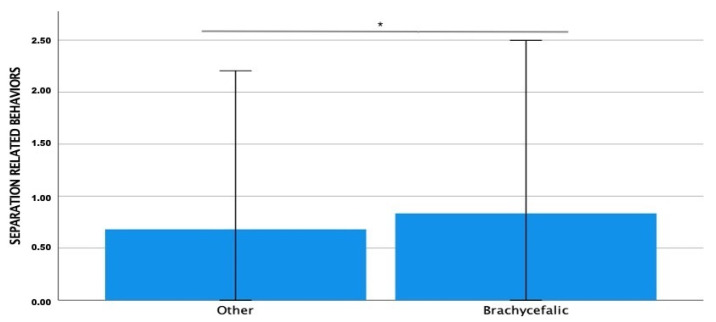
Comparison of Fe-BARQ mean scores for separation-related behaviours between brachycephalic and non-brachycephalic breeds. * *p* = 0.037.

**Figure 7 animals-13-00206-f007:**
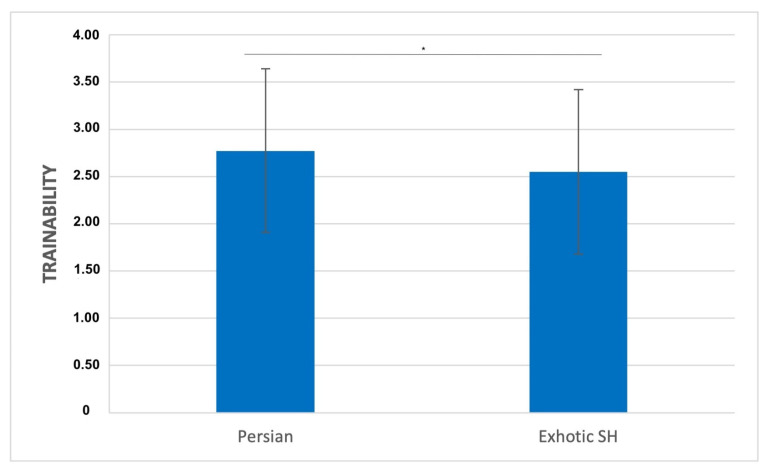
Comparison of Fe-BARQ mean scores for trainability between Persian and exotic shorthair. * *p* ≤ 0.05.

**Table 1 animals-13-00206-t001:** Representation of owners’ demographic aspects by breed (NBCOs: non-brachycephalic cat owners; BCOs: brachycephalic cat owners).

		NBCOs		BCOs	
		N°	%	N°	%
Sex	Female	216	86.4%	261	93.9%
	Male	32	12.8%	14	5.0%
	I prefer not to specify	2	0.8%	3	1.1%
Age	18–30 years	87	34.8%	41	14.7%
	31–44 years	90	36.0%	89	32.0%
	45–54 years	46	18.4%	83	29.9%
	55–64 years	19	7.6%	54	19.4%
	65–74 years	8	3.2%	10	3.6%
	>75 years	0	0.0%	1	0.4%
Environment	Apartment	27	10.8%	40	14.4%
	Apartment with balcony	120	48.0%	147	53.0%
	Semi-detached and detached house	103	41.2%	91	32.9%
Education level	Middle and secondary school	109	43.6%	159	57.2%
	Undergraduate degree	141	56.4%	119	43.0%
People in the household	1	26	10.4%	40	14.5%
	2	98	39.4%	104	37.7%
	3	68	27.3%	66	23.9%
	4	46	18.5%	59	21.4%
	5	11	4.4%	7	2.5%
Children in the household	No	162	64.8%	186	66.9%
	Yes	88	35.2%	92	33.1%
Previously owned a cat	No	32	12.8%	30	10.8%
	Yes	218	87.2%	248	89.2%

**Table 2 animals-13-00206-t002:** Representation of cat demographics by breed.

		NBCOs		BCOs	
		N°	%	N°	%
Age	Less than 3 months	2	0.8%	2	0.7%
	3–6 months	10	4.0%	17	6.1%
	6 months–1 year	15	6.0%	21	7.6%
	1–3 years	61	24.4%	91	32.7%
	4–7 years	63	25.2%	67	24.1%
	7–10 years	47	18.8%	39	14.0%
	More than 10 years	52	20.8%	41	14.7%
Sex *	Intact female	58	23.2%	72	25.9%
	Desexed female	61	24.4%	45	16.2%
	Intact male	43	17.2%	100	36.0%
	Desexed male	88	35.2%	61	21.9%
Age of adoption	Born at home	14	5.6%	14	5.0%
	Less than 2 months	42	16.8%	6	2.2%
	2–4 months	139	55.6%	160	57.6%
	More than 4 months	55	22.0%	98	35.3%
Source of adoption *	Certified breeder (recognised by Italian authority)	48	19.2%	131	47.1%
	Non-certified breeder	0	0.0%	46	16.5%
	Rescue	60	24.0%	9	3.2%
	Private	68	27.2%	81	29.1%
	Stray	68	27.2%	1	0.4%
	Other (farm, veterinarian)	6	2.4%	10	3.6%

* Significant differences are marked with an asterisk (*p* ≤ 0.05).

**Table 3 animals-13-00206-t003:** Differences in responses between NBCO and BCO (NBCO: Non-Brachycephalic Cat Owners; BCO: Brachycephalic Cat Owners).

		NBCOs		BCOs	
		N°	%	N°	%
Veterinary check *****	1 time per year	73	29.2%	62	22.3%
	2 time per year	127	50.8%	116	41.7%
	3 time for year	32	12.8%	71	25.5%
	4 time per year	12	4.8%	16	5.8%
	5 time per year	6	2.4%	13	4.7%
Brachycephalic breeds suffer more than others *	Yes	174	69.6%	152	54.7%
	No	76	30.4%	126	45.3%
Brachycephalic breeds have more problems than others	Yes	190	76.0%	204	73.4%
	No	60	24.0%	74	26.6%
If the answer to the previous question is “yes”, indicate what problems:					
Respiratory problems *	Yes	187	74.8%	137	49.3%
	No	3	1.2%	67	24.1%
	Unsure	60	24.0%	74	26.6%
Dermatological problems	Yes	68	27.2%	53	19.1%
	No	122	48.8%	151	54.3%
	Unsure	60	24.0%	74	26.6%
Ocular problems *	Yes	152	60.8%	141	50.7%
	No	38	15.2%	63	22.7%
	Unsure	60	24.0%	74	26.6%
Behavioural problems	Yes	22	8.8%	11	4.0%
	No	168	67.2%	193	69.4%
	Unsure	60	24.0%	74	26.6%
Satisfaction with veterinary expenses	Less than expected	58	23.2%	78	28.1%
	Meet expectation	128	51.2%	150	54.0%
	More than expected	64	25.6%	50	18.0%
Satisfaction with the commitment request	Less than expected	42	16.8%	53	19.1%
	Meet expectation	158	63.2%	174	62.6%
	More than expected	50	20.0%	51	18.3%
Satisfaction with the interaction (play, cuddle request, etc.)	Less than expected	32	12.8%	43	15.5%
	Meet expectation	139	55.6%	150	54.0%
	Less than expected	79	31.6%	85	30.6%
Satisfaction with the general behaviour *	Better than expected	81	32.4%	141	50.7%
	Meet expectation	119	47.6%	111	39.9%
	Worse than expected	50	20.0%	26	9.4%

* Significant differences are marked with an asterisk (*p* ≤ 0.05); veterinary check X^2^ 17.877, DF 4; *p* = 0.001; brachycephalic breeds suffer more than others X^2^ 12.411, DF 1; *p* = 0.000; respiratory problems X^2^ 66.395, DF 2; *p* = 0.000; ocular problems X^2^ 6.597, DF 2; *p* = 0.037; satisfaction with the general behaviour X^2^ 22.652, DF 2; *p* = 0.000.

**Table 4 animals-13-00206-t004:** Cats’ health information reported by BCOs.

		BCOs	
		N°	%
Symptoms during and after meal	Vomiting	46	16.5%
	Difficulty breathing	44	15.8%
	Regurgitation	44	15.8%
	Retching	41	14.7%
Symptoms during sleeping	Snoring	113	40.6%
	Changing position	38	13.6%
	Chin in elevated position	30	10.7%
	Open mouth breathing	25	8.9%
	Apnoea	10	3.5%
Other symptoms	Epiphora	193	69.4%
	Noisy breathing	117	42%
	Breathing distress after activity	44	15.8%
	Coughing	32	11.5%
	Breathing distress in hot Climatic conditions	51	18.3%
	Heat intolerance	49	17.6%
	Dystocia (only in females)	10	9.1%
Veterinary diagnosis	Dermatitis	50	17.9%
	Allergies	36	12.9%
	Corneal ulcers	34	12.2%
Conformation-related surgeries	Ocular surgery	26	9.3%%
	Odontostomatologic surgery	19	6.8%

## Data Availability

The data presented in this study are available upon request from the corresponding author.

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
