# Peer review of "Flat-Faced or Non-Flat-Faced Cats? That Is the Question"

_animals, 2023, doi:10.3390/ani13020206_

Round 1

Reviewer 1 Report

This paper is about an extremely important topic: the welfare of animals in our care. There are not many suggestion I made, and this is an important piece of work to raise the awareness of how important is to work on owners' perception of animal welfare. The physical and mental suffering of animals due to the selection of breeds with extreme phenotypical traits deserve to be known to owners and professionals and pieces of work like this one provide evidence about how important is to try a way to effectively communicate the negative welfare impact of selecting extreme traits. 

Author Response

Reviewer 1: Animals-2072405-peer-review

Dear Authors, here below my comments to your paper

We would like to thank for these comments

Line 34

Rather than ‘were not aware of these disorders’ please put ‘were not aware of the incidence of these disorders in brachycephalic cats’

We modified the text LL 31

Lines 47-49

Can you specify what registers?

We modified the text LL 45-46

Lines 112-113

Please add that this questionnaire is the CORS in here, as well.

We added the information LL 188

Lines 227-238

I think it may be better, when you mention the perception of disease risk/health problems, to specify in the texts when you report data from the entire sample (BCO+NBCO) or data from a group of owners. For example, on line 231-234, you may specify that this was the opinion of BCO+NBCO or it might be confusing to read that respiratory diseases are considered the major health problem in brachycephalic breeds (lines 231-232) and then on lines 234-235 you report that dermatological diseases were considered the main health problem in brachycephalic cats from BCO: in table 2 this statement is not supported, NBCO reported dermatological diseases being an important problem in 27.2% of answers, while BCO report 19.3%. Did I miss or misunderstood something? Also please on line 239 add what table the text is referred to.

We rephrased the text LL 545-560

Lines 384-386

It worth also mentioning the fact that not being regularly groomed can cause severe matting in long hair cats with potentially negative dermatological consequences and that Persian cats not properly trained from an early age can become aversive to being groomed with health and behaviour negative consequences.

We modified the text as suggested LL 1457-1461

Lines 412-416

Can you expand the concepts of cognitive dissonance and psychological conflict, and how these impact cat welfare?

We modified the text as suggested LL 1492-505

Reviewer 2 Report

This ms describes results of a survey into brachycephalic vs non-brachycephalic cat ownership. It is an interesting addition to the literature. However, there are some concerns I have before recommending it for publication.

Title

A little lengthy and the last part reads like a run-on. Could it be shortened somewhat?

Simple summary

Not quite ‘simple’ enough in language. A lay person cannot be expected to know what a brachycephalic breed is. This needs to be clarified.

Also, there are problems with the English, and the first sentence doesn’t make sense to this native English. The aim statement is overly wording and difficult to understand.

L21 – Lay people will also not automatically know what the CORS is, so please explain.

L25-26 – How are the findings consistent with canine research? How specifically are they similar? Many people will only read the summary and/or abstract so all of the key information needs to be presented in them.

Abstract

L27 – Explain that both Persian and Exotic shorthair are brachycephalic breeds.

L27-30 – the aim statement is very long and sounds like a run-on, much like the title. Can this be made more concise?

L30 – suggest removing ‘a questionnaire was developed’, and replace with ‘Using a questionnaire, human-cat bond and behavioural…’ .

L32 – why were health problems and motivations only explored in BCO? Wouldn’t it be useful to compare this information with NBCO? This needs further explanation.

Intro

L46-47 – the same wording as in the summary and it’s unclear, especially the bit ‘globally assisting to’. This doesn’t make sense.

L50 – cite the claim that Persian and Exotics are the most popular cat breeds in the world, and clarify that ‘domestic shorthair’ isn’t technically a breed, because they are almost certainly the most common kind of cat in the world.

 L52 – do the authors mean to say ‘exaggerated’ instead of ‘exasperated’?

L65-67 – cite this claim

L85-88 – this is a strong sentence, but it shouldn’t be its own paragraph. Suggest adding it to the end of the previous one or the beginning of the next one.

L107 – suggest changing ‘the last aim’ to ‘a secondary aim’

Methods

Section 2.1. – suggest making the full survey available as supp mat for anyone who wants to replicate the study down the track. Since some of the scales are already published, the supp mat can include a reference to the original publication instead of the scale itself, where relevant.

L113 – suggest changing from ‘developed’ to ‘created’, because survey development has a specific definition and it involves many steps.

L114 – it’s not clear when these are referring to owners and when they’re referring to cats. It’s also not clear what ‘motivations’ and ‘perceptions’ mean because they are very broad terms.

L124 – please explain why these questions weren’t also included for BCOs. It would be an interesting comparison. As it stands, the results will only allow an understanding of how often these problems are occurring in brachycephalic cats, but without any reference from non-brachcephalic cats.

L132 – Emotional Closeness should be Perceived Emotional Closeness, according to the original CORS and CDORS.

L155 – cite the fe-BARQ

L156 – don’t assume the reader knows what the C-BARQ is. Clarify. Also, cite the C-BARQ here. Both citations are listed at the bottom of the para but it should be clear which citation relates to which scale.

L174-175 – since the data were collected during COVID restrictions, an unusual time in the lives of the cats and their owners, this should be mentioned in the Discussion section somewhere: what are the implications of collecting this data during the pandemic? It could have affected the CORS results if people are spending more time than usual with their cats. It could also have affected the fe-BARQ results because people may have noticed for the first time that their cat was behaving in problematic ways, or perhaps the cat behaved better than usual because the owner was always around. This needs to be a para in the Discussion.

Methods – there’s not enough info on the participants. What language was the survey in? Was there a minimum owner age for participation? What about a minimum cat age/time living with owner? What were the inclusion/exclusion criteria?

L186 – there’s a random ‘p’ toward the end of this line.

L189 – how were the CORS subscales created? Were they averaged or summed? What’s the total possible range for each subscale? This is important to give context to descriptive results.

Results

L207 – these should be cat owners, not cats, and suggest adding NBCO and BCO to each to remind the reader what they mean

L214 – suggest changing ‘demographic data by breeds’ to ‘demographic data among NBCO vs BCO …’

Tables 1 and 2 – suggest writing out the meaning of BCO and NBCO in the caption or as a footnote. Tables and figures should be able to stand alone without requiring the reader to look for further detail in the text, so if someone doesn’t want to read the text, they should still be able to understand what BCO and NBCO represent.

Tables 1 and 2 – suggest reordering Age and Education level so that the lowest comes first (e.g., the Age shouldn’t start with >75 years, it should start with 18-30 and >75 should be the last listed).  Also for table 2, the source of adoption should list ‘other’ as the last one, and a footnote would be useful to explain what ‘other’ meant in some cases.

Tables 1 and 2 – in international journals, usually a . is used in decimals instead of a , This is done mostly correctly in text but not in the tables.

Tables 1 and 2 – could statistical results be added as well, since x2 tests were used to investigate differences?

Table 2 – suggest changing ‘female’ and ‘male’ to ‘entire female’ and ‘entire male’ in cat sex.

Table 2 - ‘certificated’ should be ‘certified’, but it’s not clear what this means. Please describe.

L228 – where was this question mentioned in the methods section? Please clarify in the methods what sorts of questions were asked about this. This is also why a supp mat with the full survey would be useful.

L228-230 – it would make more sense to compare NBCO and BCO, rather than BCO with the total sample. The way it is currently written, the data are not independent.

L231-234 – for all owners or BCO only?

L234 – provide the numbers for this result

L235-237 – this is written very awkwardly. It could be rephrased for simplicity and clarity.

Table 3 – Could the statistical results be put into this table as well, including x2 and any post-hoc tests?

Table 3 – ‘Time for year’ should be ‘time per year’

Table 3 – Suggest changing ‘yes/no/unsure’ items to order as ‘yes’, ‘no’, and ‘unsure’, rather than the other way around.

Table 3 – satisfaction with behaviour, suggest changing response order to less, meet, more, as in the other items

L243 – the first part of this sentence is worded awkwardly

L243-267 – suggest putting this info in a table because these are key results and the way they are currently formatted makes them difficult to read. The reader may skim over the results and miss important findings this way.

L269 – but a higher score on PC means bad, right? As in more perceived costs? It’s generally recommended that items be reverse scored as required so that a higher score on PC indicates better relationship, and lower perceived costs. The authors appear not to have done this, as they described in the methods. Please clarify.

Figs 1-3 – In the captions, please clarify whether a high score is better or worse.

L281-287 – these are not the full regression results. More than the p-value needs to be presented here. See https://www.statology.org/how-to-report-regression-results/

Section 3.4 – were the x2 tests? If so, please report the x2 statistic as well as the p-value. If not, explain what test was done and report the full results. I also recommend, for all analyses, to consider doing an effect size analysis, if possible, that will tell the reader whether the results are meaningful, regardless of their p-value. That will also help everyone understand whether a ‘tendency’ of p = 0.08 has any potential meaning.

Discussion

L324-325 – unclear wording

L327 – what made the questionnaire ‘ad hoc’?

L327-329 – this sentence is unnecessarily wordy. Suggest rephrasing to ‘Using an online questionnaire, we investigated bracycephelic and non-brachycephalic cat owners’ motivation for purchasing cats and perception of their cats’ health and welfare, as well as cat behaviour and the human-cat bond.’

L355 – relation should be relationship

L350-351 – this sentence is awkwardly phrased

L352-353 – why might this be the case? Also, consider whether this information really adds anything to the main story. Do these minor demographic differences really affect the overall picture? If not, they can probably be removed from the Discussion. Maybe just pick one or two of the unexpected demographic results to focus on in the Discussion.

L363 – ‘subject’ is a strange word to use here.

L365 – less likely than whom?

L387 – no sig differences between groups in what context? Explain

L440 – a sentence should not be a para. L440-452 could all be one para.

L506 and elsewhere – i.e. should be e.g. ie = that is; eg = for example

Limitations para – these limitations are pretty pedestrian and would apply to most cross-sectional survey studies. It would be better to focus on limitations that are specific to this study (e.g., mostly women, COVID data collection).

Author Response

Comments and Suggestions for Authors

This ms describes results of a survey into brachycephalic vs non-brachycephalic cat ownership. It is an interesting addition to the literature. However, there are some concerns I have before recommending it for publication.

We would like to thank the reviewer for the valuable comments to improve our manuscript to be published

Title

A little lengthy and the last part reads like a run-on. Could it be shortened somewhat?

We modified the title according to the suggestion

Simple summary

Not quite ‘simple’ enough in language. A lay person cannot be expected to know what a brachycephalic breed is. This needs to be clarified.

Also, there are problems with the English, and the first sentence doesn’t make sense to this native English. The aim statement is overly wording and difficult to understand.

L21 – Lay people will also not automatically know what the CORS is, so please explain.

L25-26 – How are the findings consistent with canine research? How specifically are they similar? Many people will only read the summary and/or abstract so all of the key information needs to be presented in them.

We modified the simple summary as suggested

Abstract

L27 – Explain that both Persian and Exotic shorthair are brachycephalic breeds.

We modified the text LL 24

L27-30 – the aim statement is very long and sounds like a run-on, much like the title. Can this be made more concise?

We modified the text LL 25-27

L30 – suggest removing ‘a questionnaire was developed’, and replace with ‘Using a questionnaire, human-cat bond and behavioural…’ .

We rephrased the sentence LL 27

L32 – why were health problems and motivations only explored in BCO? Wouldn’t it be useful to compare this information with NBCO? This needs further explanation.

Thanks for your comment, it’s a very interesting point. We wanted to focus on the clinical issues of brachycephalic as genetically maltreated breeds, but surely, we will deepen this point in further studies.

Intro

L46-47 – the same wording as in the summary and it’s unclear, especially the bit ‘globally assisting to’. This doesn’t make sense.

We rephrased the sentence LL 43-44

L50 – cite the claim that Persian and Exotics are the most popular cat breeds in the world, and clarify that ‘domestic shorthair’ isn’t technically a breed, because they are almost certainly the most common kind of cat in the world.

We added the reference as required. We didn’t mention “domestic shorthair”; British shorthair and Exotic shorthair are recognized breeds. LL 116-117

 L52 – do the authors mean to say ‘exaggerated’ instead of ‘exasperated’?

We modified the text LL 119

L65-67 – cite this claim

We added the references LL 134

L85-88 – this is a strong sentence, but it shouldn’t be its own paragraph. Suggest adding it to the end of the previous one or the beginning of the next one.

We modified the text LL 150-154

L107 – suggest changing ‘the last aim’ to ‘a secondary aim’

We modified the text as suggested LL 180

Methods

Section 2.1. – suggest making the full survey available as supp mat for anyone who wants to replicate the study down the track. Since some of the scales are already published, the supp mat can include a reference to the original publication instead of the scale itself, where relevant.

Thank you for your comment. The questionnaire is very long, so we are available to send it to those who request it directly.

L113 – suggest changing from ‘developed’ to ‘created’, because survey development has a specific definition and it involves many steps.

We modified the text LL 186

L114 – it’s not clear when these are referring to owners and when they’re referring to cats. It’s also not clear what ‘motivations’ and ‘perceptions’ mean because they are very broad terms.

We modified the text to clarify the text. LL 185-189

L124 – please explain why these questions weren’t also included for BCOs. It would be an interesting comparison. As it stands, the results will only allow an understanding of how often these problems are occurring in brachycephalic cats, but without any reference from non-brachcephalic cats.

Thanks for your comment. As said previously, this is a very interesting issue. We wanted to focus on the clinical issues of brachycephalic as genetic maltreatment breeds, but surely, we will deepen this point in further studies.

L132 – Emotional Closeness should be Perceived Emotional Closeness, according to the original CORS and CDORS.

We corrected the text LL207

L155 – cite the fe-BARQ

We corrected the text LL 280-284

L156 – don’t assume the reader knows what the C-BARQ is. Clarify. Also, cite the C-BARQ here. Both citations are listed at the bottom of the para but it should be clear which citation relates to which scale.

We modified the text LL 280-288

L174-175 – since the data were collected during COVID restrictions, an unusual time in the lives of the cats and their owners, this should be mentioned in the Discussion section somewhere: what are the implications of collecting this data during the pandemic? It could have affected the CORS results if people are spending more time than usual with their cats. It could also have affected the fe-BARQ results because people may have noticed for the first time that their cat was behaving in problematic ways, or perhaps the cat behaved better than usual because the owner was always around. This needs to be a para in the Discussion.

We modified the text as suggested in Discussion LL 1745-1749; LL 1991-1994

Methods – there’s not enough info on the participants. What language was the survey in? Was there a minimum owner age for participation? What about a minimum cat age/time living with owner? What were the inclusion/exclusion criteria?

We added the information required in the text (LL 301-309). We didn’t ask for minimum cat age/time living with owner. We collected age of adoption and age of the cat and we put this information in the result section (table 2).

L186 – there’s a random ‘p’ toward the end of this line.

Corrected

L189 – how were the CORS subscales created? Were they averaged or summed? What’s the total possible range for each subscale? This is important to give context to descriptive results.

We followed the scoring instructions for the C/DORS realized by the authors (Howell et al., 2016). Each item is scored on a five-point scale from 1 to 5. Items in the Pet-Owner interaction and Perceived Emotional Closeness subscales were be reversed scored, such that a higher score indicates better perceived relationship quality. To calculate the score for each subject for each subscale we summed the scores and then divided by the number of items of the subscale; then we calculated the subscale average for each group (BCO and NBCO). We added the information in the text (LL 220-225)

Results

L207 – these should be cat owners, not cats, and suggest adding NBCO and BCO to each to remind the reader what they mean

Corrected LL 342-343

L214 – suggest changing ‘demographic data by breeds’ to ‘demographic data among NBCO vs BCO …’

Corrected LL 349-350

Tables 1 and 2 – suggest writing out the meaning of BCO and NBCO in the caption or as a footnote. Tables and figures should be able to stand alone without requiring the reader to look for further detail in the text, so if someone doesn’t want to read the text, they should still be able to understand what BCO and NBCO represent.

Thanks for your suggestion: we added the information in the captions.

Tables 1 and 2 – suggest reordering Age and Education level so that the lowest comes first (e.g., the Age shouldn’t start with >75 years, it should start with 18-30 and >75 should be the last listed).  Also for table 2, the source of adoption should list ‘other’ as the last one, and a footnote would be useful to explain what ‘other’ meant in some cases.

We modified the tables according to the comments

Tables 1 and 2 – in international journals, usually a . is used in decimals instead of a , This is done mostly correctly in text but not in the tables.

We modified the tables according to the comments

Tables 1 and 2 – could statistical results be added as well, since x2 tests were used to investigate differences?

Thanks for your comment. The results reported in table 1 were not statistically significant. We added the significance in table 2.

Table 2 – suggest changing ‘female’ and ‘male’ to ‘entire female’ and ‘entire male’ in cat sex.

We modified the table according to the comments

Table 2 - ‘certificated’ should be ‘certified’, but it’s not clear what this means. Please describe.

We modified the table according to the comments

L228 – where was this question mentioned in the methods section? Please clarify in the methods what sorts of questions were asked about this. This is also why a supp mat with the full survey would be useful.

We added an explanation in M&M section LL 198-200

L228-230 – it would make more sense to compare NBCO and BCO, rather than BCO with the total sample. The way it is currently written, the data are not independent.

We modified the text LL 545-560

L231-234 – for all owners or BCO only?

We modified the text LL 545-560

L234 – provide the numbers for this result

We modified the text LL 545-560

L235-237 – this is written very awkwardly. It could be rephrased for simplicity and clarity.

We modified the text LL 545-560

Table 3 – Could the statistical results be put into this table as well, including x2 and any post-hoc tests?

We added the information requested in the table

Table 3 – ‘Time for year’ should be ‘time per year’

We modified the table according to the comment

Table 3 – Suggest changing ‘yes/no/unsure’ items to order as ‘yes’, ‘no’, and ‘unsure’, rather than the other way around.

We modified the table according to the comments

Table 3 – satisfaction with behaviour, suggest changing response order to less, meet, more, as in the other items

We use the same approach of Plitman et al., 2019

L243 – the first part of this sentence is worded awkwardly

We modified the text LL 750

L243-267 – suggest putting this info in a table because these are key results and the way they are currently formatted makes them difficult to read. The reader may skim over the results and miss important findings this way.

Thanks for this suggestion, we modified the paper accordingly we added table 4

L269 – but a higher score on PC means bad, right? As in more perceived costs? It’s generally recommended that items be reverse scored as required so that a higher score on PC indicates better relationship, and lower perceived costs. The authors appear not to have done this, as they described in the methods. Please clarify.

We added the information in M&M section

Figs 1-3 – In the captions, please clarify whether a high score is better or worse.

We added the explanation in M&M section LL202-225

L281-287 – these are not the full regression results. More than the p-value needs to be presented here. See https://www.statology.org/how-to-report-regression-results/

We added the information requested in the text LL1150-1153

Section 3.4 – were the x2 tests? If so, please report the x2 statistic as well as the p-value. If not, explain what test was done and report the full results. I also recommend, for all analyses, to consider doing an effect size analysis, if possible, that will tell the reader whether the results are meaningful, regardless of their p-value. That will also help everyone understand whether a ‘tendency’ of p = 0.08 has any potential meaning.

The data of Fe-BARQ were not normally distributed so a Kruskal-Wallis equality of population rank test was used to detect the presence of a significant association between Fe-BARQ factor score and cat demographic (breed, age, sex, age and source of adoption); cat clinical data; owner demographics (sex, family members); veterinary routine; owner expectation of breed versus reality ownership (veterinary cost, activity level, overall behaviour). Mann-Whitney U Test for non-normally distributed continuous * categorical data (Fe-BARQ score * breed) was performed.

We added further statistical results (section 3.4) LL 1161-1163; LL1171-1172; LL 1191-1196

Discussion

L324-325 – unclear wording

We rephrased the sentence LL1201-1204

L327 – what made the questionnaire ‘ad hoc’?

We rephrased the sentence LL 1204-1206

L327-329 – this sentence is unnecessarily wordy. Suggest rephrasing to ‘Using an online questionnaire, we investigated bracycephelic and non-brachycephalic cat owners’ motivation for purchasing cats and perception of their cats’ health and welfare, as well as cat behaviour and the human-cat bond.’

We rephrased the sentence LL 1204

L355 – relation should be relationship

We modified the text

L350-351 – this sentence is awkwardly phrased

We rephrased the sentence LL 1280-1281

L352-353 – why might this be the case? Also, consider whether this information really adds anything to the main story. Do these minor demographic differences really affect the overall picture? If not, they can probably be removed from the Discussion. Maybe just pick one or two of the unexpected demographic results to focus on in the Discussion.

We rephrased the sentence LL 1281-1286

L363 – ‘subject’ is a strange word to use here.

We modified the text LL 1293

L365 – less likely than whom?

Respect to owners of other breeds

L387 – no sig differences between groups in what context? Explain

In this study LL 1462

L440 – a sentence should not be a para. L440-452 could all be one para.

We modified the text LL 1735

L506 and elsewhere – i.e. should be e.g. ie = that is; eg = for example

Corrected LL 1982

Limitations para – these limitations are pretty pedestrian and would apply to most cross-sectional survey studies. It would be better to focus on limitations that are specific to this study (e.g., mostly women, COVID data collection).

We modified the text as suggested LL 2113-2130
